# A modification of technology acceptance model for investigating driver-vehicle interaction systems usage

**Yichen Dong** [1☉*], **Makoto Itoh** [2,3☉]

**1** Degree Program of Risk and Resilience Engineering, Graduate School of System and Information Engineering, University of Tsukuba, Tsukuba, Ibaraki, Japan, **2** Institute of Systems and Information Engineering, University of Tsukuba, Tsukuba, Ibaraki, Japan, **3** Center for Artificial Intelligence Research, University of Tsukuba, Tsukuba, Ibaraki, Japan

☉ These authors contributed equally to this work.
* dong@css.risk.tsukuba.ac.jp

**Data availability statement:** All questionnaire origin data are

## Abstract

This descriptive study investigated respondents' acceptance of driver-vehicle interaction systems by modifying the technology acceptance model using user experience concepts. A questionnaire survey was conducted, which including 15 variables and a Likert scale (1–5) range from 'strongly agree' to 'strongly disagree' was adopted for all variables. The questionnaire examined six constructs proposed in the modified technology acceptance model: perceived usefulness, perceived enjoyment, satisfaction, attitude, interactive media, and user interface. 495 samples aged from 21 to 82 years old (48.7 ± 12.5 years) including 279 males and 216 females were collected for data analysis. Confirmatory factor analysis demonstrated the reliability and validity of the model using several indices. Correlations between constructs were proven using path analysis ($p < 0.05$). Then, the influences of drivers' gender (male = 1 and female = 2) and age on the constructs were analyzed using linear regression analysis. Male drivers had higher perceived usefulness (Beta = 0.235, $p < 0.01$) and more positive attitudes (Beta = 0.087, $p < 0.01$) than female drivers. However, their perceived enjoyment of the system was lower than that of females (Beta = -0.135, $p < 0.01$). Older age led drivers to prefer a negative attitude toward driver-vehicle interaction systems (Beta = 0.072, $p < 0.01$) and dislike utilizing the user interface (Beta = 0.139, $p < 0.01$). Finally, four implications based on the analysis were proposed to guide the design of driver-vehicle interaction systems. This study clarified the feasibility of the modified technology acceptance model for investigating Japanese driver's opinions on driver-vehicle interaction systems and provided insights for designing vehicle human-machine interfaces to improve driver acceptance.

## Introduction

Intersection research on technology, policy, and human behavior in transportation seeks to decrease road accidents, injuries, and deaths by minimizing human errors and enhancing road design. One focus of this study is the implementation of advanced vehicle automation

available from the Dryad database (http://datadryad.org/share/FdQI7YfqFsVyzv-MnmHGvXvAsiObJNPcYAOTC2sCwi4). Forthcoming on Dryad. https://doi.org/10.5061/dryad.4qrfj6qm5.

**Funding:** This study received the research budget from the JSPS KAKENHI (Japan Society for the Promotion of Science Grants-in-Aid for Scientific Research), MI received, and the JST SPRING (Japan Science and Technology Agency Support for Pioneering Research Initiated by the Next Generation), YD received. The grant number of JSPS KAKENHI is 24H00361, and the URL is https://www.jsps.go.jp/english/egrants/index.html The grant number of JST SPRING is JPMJSP2124, and the URL is https://www.jst.go.jp/jisedai/spring/en/index.html The sponsor didn't play any role in the study.

**Competing interests:** The authors have declared that no competing interests exist.

technologies that consider both safety and user acceptance. However, the public still has doubts and concerns about vehicle automation owing to the potential risks of driving automated vehicles and the driver's bias toward perceived safety [1]. The solution is to let the vehicle manufacturer clearly state the vehicle automation scenario with a high level (i.e., level 3 or higher) to the public to decrease misunderstandings and boost their confidence in vehicle automation technologies [2,3].

In high-level vehicle automation scenarios, drivers do not need to constantly monitor vehicle situations. Hence, it is becoming possible for drivers to execute non-driving-related tasks (NDRTs) in the car, such as playing a game or watching a movie. To satisfy drivers' requirements for conducting NDRTs, vehicle manufacturers have developed human-machine interaction (HMI) systems between drivers and autonomous vehicles [4]. Regarding research on vehicle HMI, the number of articles was low. However, the steady increase in vehicle HMI articles over the past 20 years reflects the growing importance of technologies and methodologies for in-vehicle interactions [5].

The concept of the driver-vehicle interaction system (DVIS) was proposed to study vehicle HMI. It is an information system that integrates different technologies, such as haptic, visual, and auditory devices, and is expected to provide drivers with natural, friendly, and adaptive interactions [6]. One example is 'Blue&Me' innovative telematics solution by FIAT (S1 File). Detjen et al. [7] suggested that vehicle interaction systems should be acceptable to drivers to encourage their adoption. One possible method for investigating driver acceptance is to introduce technology acceptance model (TAM) [8].

In this study, we propose a modified TAM with six constructs to investigate drivers' acceptance of the DVIS and prove construct correlations. In addition, the influence of drivers' gender and age on the constructs is clarified.

## Materials and methods

### TAM

Davis [8] proposed TAM to predict an individual's perceived adoption of information technology. TAM is recognized as the most famous and widely used model of rational behavior. There are three determinant constructs of TAM, as illustrated in Fig 1. The intention to use is influenced by two constructs in the model: perceived usefulness and perceived ease of use.

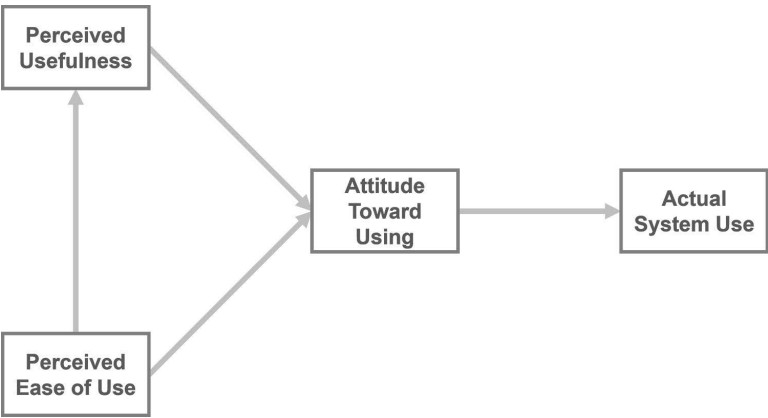

**Fig 1. Technology acceptance model [8].**

Perceived ease of use directly influences behavioral intention in the same manner as does perceived usefulness. Simultaneously, this affects perceived usefulness.

Regarding the application of TAM, system engineers and designers adopt it to control the degree of user perception of systems and investigate user preferences. Related study have used TAM to analyze user perceptions of vehicle systems [9]. However, adopting TAM for predicting driver acceptance of the DVIS faces several challenges. First, perceived usefulness is connected to task performance [10], which is unsuitable for researching the DVIS because the process of using the DVIS is not considered a task. Second, the investigative objective of TAM is engineers, workers, and students [11], which is different from that of the DVIS. In general, a driver is no longer considered to be an occupation. Third, existing automated systems studied by TAM have experienced a significant division between discretionary and mandatory utilization. It is necessary to clarify that TAM is suitable for adoption in discretionary systems (i.e., vehicle systems) [12]. Finally, the interaction environment of the vehicle differs from that of the original information systems. Unlike personal computers, drivers can use these systems on moving vehicles at any time.

Owing to these challenges, the original TAM is unsuitable for investigating the DVIS. TAM should be modified specifically for the DVIS. One possible method is to modify TAM using user experience (UX) knowledge. UX is the subject of research on user experience when interacting with a product, system, or service. This includes user perception of utility, ease of use, and efficiency. The method for studying UX is to develop a related model that discusses user experience when utilizing an interaction system. One characteristic of the UX model is hedonic. It describes how exciting and stimulating a system provides a user with an experience [13].

Some evidence supports the modification of the TAM with the UX model. First, UX has become an important topic in the HMI research field because it focuses on the user aspect to consider how to enhance interactive processes effectively [14]. The research target of UX is the same as that of TAM, which investigates users rather than machines. Portz et al. [15] suggested that users prefer to accept a technological system with good UX design. Finally, Hornbæk and Hertzum [16] considered the hedonic characteristics to be significant in connecting UX and TAM constructs. Hedonic characteristics actively explain how user motivation can be improved using a system.

Several studies have connected the UX model to TAM [13,15]. The connecting method is to introduce factors in hedonic characteristics as constructs of TAM. For example, Ghazizadeh et al. [12] adapted hedonic characteristics to predict perceived ease of use.

Therefore, combining the UX model with TAM should extend the usage of TAM in the DVIS, which will contribute to predicting whether drivers will adopt the DVIS in the future, and motivate the development of the UX and TAM research fields.

## Proposed model

We propose a modified TAM by adopting the UX concept, including six constructs for investigating the driver's acceptance of the DVIS: perceived usefulness, perceived enjoyment, satisfaction, attitude, interactive media, and user interface.

## Perceived usefulness

Perceived usefulness is the degree to which a person considers utilizing a particular system to improve job performance [10]. Mustafa et al. [17] discovered that perceived usefulness contributes positively to customer perception of vehicle adoptions. Thus, we retain the construct of perceived usefulness in our proposed model. However, the original TAM

questionnaire is not satisfactory for research on the DVIS because some functions in the DVIS represent the hedonic aspect, such as climate control and audio/video. Previous study alternated TAM questionnaire wording to focus on vehicle aspects [18]. In our previous study [19], a questionnaire was designed to investigate drivers' perceived usefulness of the information in a DVIS. We followed our previous questionnaire wording and described perceived usefulness as a driver's ability to identify and understand the information provided by the DVIS.

### Perceived enjoyment

One hedonic factor in modifying TAM by introducing the UX concept is perceived enjoyment [20,21]. Beggiato et al. [22] found that enjoyment is a significant factor for drivers to accept autonomous vehicles. In addition, perceived enjoyment influences customer's perceptions of the vehicle before making their decisions for adopting [23]. Thus, we introduce perceived enjoyment as one construct of the proposed model and define it as the comfort provided by the DVIS interaction functions perceived by drivers.

Previous studies argued that perceived enjoyment is strongly connected with perceived ease of use [20,24]. In this study, we propose that perceived enjoyment should replace perceived ease of use. Related studies have supported this replacement [25].

Regarding the correlation of perceived enjoyment, we hypothesize that perceived usefulness positively influences perceived enjoyment of using a DVIS, as supported by [26].

### Satisfaction

In usability engineering, satisfaction is defined as a user's hedonic value when using a system [27]. In this study, we define satisfaction as the total evaluation of the driver's consideration of the enjoyment level of the DVIS. The relationship between user perceptions of information system adoption and satisfaction has been examined in several studies [28–30]. Therefore, it is possible to add satisfaction as a construct to TAM. Related studies support this possibility [31,32].

Regarding the relationship between satisfaction and other TAM constructs, satisfaction is correlated with perceived usefulness and perceived ease of use [33–35]. However, we substitute perceived enjoyment for perceived ease of use. Thus, we hypothesize that satisfaction is positively influenced by perceived enjoyment. In addition, this study considers that satisfaction directly affects perceived usefulness. This is because, if a system satisfies users' expectations, users will improve their understanding of it compared with other systems with a low level of satisfaction.

### Attitude

Attitude toward using is a determinant for a user to utilize or reject a system, and it is influenced by perceived ease of use and perceived usefulness [8]. Davis et al. [36] modified TAM by replacing attitude with behavioral intention. However, some researchers have argued that attitude is more significant than behavioral intention in explaining systems' hedonic aspects, and that the cognitive and affective perspective variables of attitude should be considered in TAM [16,37]. Based on these suggestions, we maintained the attitude construct in the proposed model and designed two questions to explain the cognitive and affective variables of attitude. The questions are presented in S1 Table (i.e., AT1 and AT2).

Regarding the relationship between attitude and other constructs in TAM, we follow the suggestions in [8]. We then replace the correlation between perceived ease of use with perceived enjoyment, which is supported by [16].

## System characteristics

Actual system use is treated as a response in TAM, which is directly influenced by users' attitudes [8]. This construct is typically defined as an objective system that has been investigated previously. However, Hornbæk and Hertzum [16] suggested that it is necessary to verify the system characteristics of TAM. To determine the characteristics of the DVIS, we introduced the AT-ONE method [38], which inspires designers to develop an interaction process for systems. Considering the interaction process between users and the DVIS concluded by the AT-ONE method, the DVIS is divided into two characteristics: interactive media and user interface. These characteristics are described below.

Interactive media: a driver requires a touch point to interact with the DVIS. Therefore, they should initially select interactive media to input their ideas and perceptions into the system.

User interface: after receiving the driver's intention, the system provides information related to the driver through a user interface.

Regarding the correlations of these two characteristics, we follow [8] and hypothesize that attitude influences interactive media and user interface.

Finally, Table 1 summarizes the description of each construct in the proposed model.

## Demography

User demographics are significantly related to perceptions and attitudes toward utilizing specific systems or technologies [39]. In addition, user demographics influence driver's UX scores for using driver assistance systems [31]. Thus, driver demographics (e.g., gender and age) are considered influencing variables for the constructs in the proposed model. We hypothesize that the driver's gender and age influence each construct in the proposed model (i.e., perceived usefulness, perceived enjoyment, satisfaction, attitude, interactive media, and user interface).

## Hypotheses

Table 2 summarizes the hypotheses from the previous discussion. In addition, the proposed model based on explanations of the constructs and their correlation hypotheses is illustrated in Fig 2.

## Methodology

### Sample

The target respondents of this study are Japanese people with driving licenses, meaning that only a person with experience in utilizing vehicle systems is included in the study. A total of 495 respondents' opinions were collected and used for data analysis (N = 495). Male = 279 and

**Table 1. Descriptions of the constructs in the proposed model.**

|     | Definition | Reference |
| --- | --- | --- |
| PU | A driver's ability to identify and understand information provided by the DVIS. | [18,19] |
| PE | The comfort provided by the DVIS interaction functions is perceived by drivers. | [25,26] |
| ST | The total evaluation of the driver's consideration of the enjoyment level of the DVIS. | [31–35] |
| AT | A determinant for a driver to utilize or reject the DVIS. | [8,16,37] |
| IM | The touchpoint that a driver interacts with the DVIS. | [8,16] |
| UI | The interface that the DVIS provides information for a driver. | [8,16] |

PU, perceived usefulness; PE, perceived enjoyment; ST, satisfaction; AT, attitude; IM, interactive media; UI, user interface.

**Table 2. Descriptions of the hypotheses.**

| | |
|---|---|
| H1 | Perceived usefulness has a positive effect on perceived enjoyment of utilizing DVIS. |
| H2 | Satisfaction has a positive effect on perceived usefulness of utilizing DVIS. |
| H3 | Perceived enjoyment has a positive effect on satisfaction of utilizing DVIS. |
| H4 | Perceived usefulness has a positive effect on drivers' attitude toward using DVIS. |
| H5 | Perceived enjoyment has a positive effect on drivers' attitude toward using DVIS. |
| H6 | Driver's attitude toward using DVIS has a positive effect on the use of interactive media. |
| H7 | Driver's attitude toward using DVIS has a positive effect on the use of user interface. |
| H8 | Driver's gender has significant relationships with each construct in the research model. |
| H9 | Driver's age has significant relationships with each construct in the research model. |

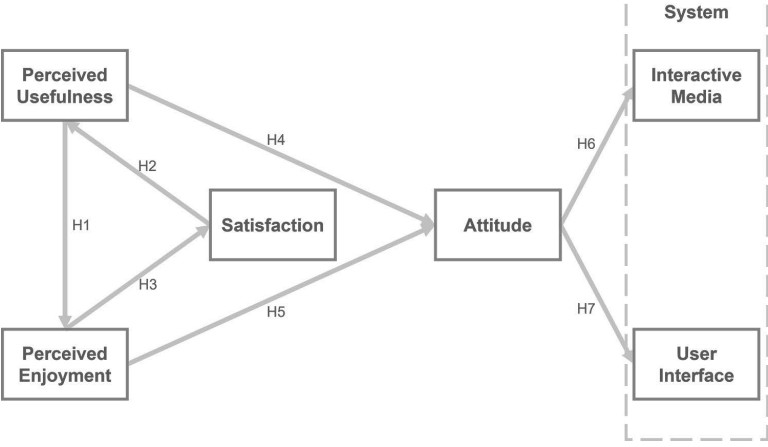

**Fig 2. Proposed model with construct correlation hypotheses.**

female = 216, respondents' ages ranged from 21 to 82 years (Mean = 48.7, Standard deviation = 12.5).

## Questionnaire design

We used a questionnaire survey method to collect respondents' opinions. A questionnaire with 15 variables was designed to present the proposed model, and a Likert scale (1–5) range from 'strongly agree' to 'strongly disagree' was adopted for all variables. The variables of the questionnaire are presented in S1 Table.

The questionnaire was modified from our previous work [19], which was proven to be reliable and valid for investigating Chinese perceptions of using the DVIS by exploratory factor analysis (Cronbach's Alpha range from 0.633 to 0.849, Corrected Item-Total Correlation values > 0.4, KMO = 0.776, Bartlett's Test of Sphericity was significant). A bilingualist translated and checked the questionnaire from Chinese into Japanese.

This study was approved by the ethical committee of the Institute of System and Information Engineering, the University of Tsukuba. The approval number is 2021R551-1. Respondents with driving licenses and ages from 20 to 99 can complete the questionnaire. The questionnaire was distributed by the Rakuten Survey Company, which specializes in online surveys. The company has a fixed respondent group for the questionnaire survey, who are hired by this company. Hence, the distribution period consisted of two days, from 1/Sep/2022 to 2/Sep/2022.

**Table 3. Descriptive statistics of variables.**

|  | Mean | Standard deviation | Skewness | Kurtosis |
|---|---|---|---|---|
| PU |  |  |  |  |
| PU1 | 3.26 | 1.13 | -0.186 | -0.702 |
| PU2 | 3.37 | 1.09 | -0.168 | -0.671 |
| PU3 | 3.29 | 1.14 | -0.187 | -0.605 |
| PU4 | 3.53 | 1.02 | -0.236 | -0.187 |
| PE |  |  |  |  |
| PE1 | 2.78 | 1.03 | 0.447 | -0.082 |
| PE2 | 3.09 | 1.00 | 0.228 | -0.224 |
| ST |  |  |  |  |
| ST1 | 2.86 | 1.06 | 0.277 | -0.272 |
| ST2 | 2.98 | 0.96 | 0.286 | 0.213 |
| AT |  |  |  |  |
| AT1 | 3.28 | 1.00 | 0.100 | -0.331 |
| AT2 | 3.58 | 1.05 | -0.260 | -0.477 |
| IM |  |  |  |  |
| IM1 | 3.27 | 1.25 | -0.049 | -1.028 |
| IM2 | 3.87 | 1.07 | -0.620 | -0.407 |
| IM3 | 3.60 | 1.15 | -0.256 | -0.935 |
| UI |  |  |  |  |
| UI1 | 2.50 | 1.14 | 0.689 | -0.175 |
| UI2 | 2.77 | 1.08 | 0.462 | -0.306 |

PU, perceived usefulness; PE, perceived enjoyment; ST, satisfaction; AT, attitude; IM, interactive media; UI, user interface.

## Measurement and data analysis

First, we evaluated the respondents' data to delete invalid data. The standard of deletion is that a respondent had answered with the same level of over 70% of the variables in the question-naire. None of the data was deleted with the standard. Then, we tested the normality of the variables. The descriptive statistics for the variables are shown in Table 3. From the table, all the variables are suitable for the normality distribution (absolute value of skewness < 10 and kurtosis < 3).

IBM SPSS Statistics (version 27.0.1.0) and IBM SPSS Amos (version 26.0.0) were used for statistical analyses. Differences were considered statistically significant at $p < 0.05$. We divided the analysis processes separately to prove the feasibility of the proposed model and its con-structs. Initially, a confirmatory factor analysis was used to check the reliability and validity of the model. Then, path analysis clarified the correlations between the constructs in the model. Finally, linear regression analyses were conducted to investigate the influence between demo-graphics and constructs.

## Results

Table 4 lists the several indices used to estimate the fitness of the measurement model. First, owing to the large sample size of the data, the ratio of chi-square to degrees of freedom replaced the chi-square statistic (chi-square = 273.940, degrees of freedom = 83). Its value is 3.30, which is unsuitable because the standard value is < 3.0. However, the goodness-of-fit (GFI) and adjusted goodness-of-fit (AGFI) indicate a good fit for the model (GFI = 0.932 and

**Table 4. Fit indices for measurement model.**

| Fit Indices | Recommended value | Measurement model |
|---|---|---|
| Chi-square/ degrees of freedom | ≤ 3.00 | 3.30 |
| GFI | ≥ 0.900 | 0.932 |
| AGFI | ≥ 0.800 | 0.901 |
| NFI | ≥ 0.900 | 0.945 |
| NNFI | ≥ 0.900 | 0.950 |
| CFI | ≥ 0.900 | 0.961 |
| RMSR | ≤ 0.100 | 0.058 |
| RMSEA | ≤ 0.080 | 0.068 |

AGFI = 0.901). The comparative fit index (CFI), normalized fit index (NFI), and non-normal fit index (NNFI) were used for validation. Typically, to represent the fitness of a model, these values should be greater than 0.9 to represent the fitness of the model. From our observations in the measurement model, CFI, NFI, and NNFI are 0.961, 0.945, and 0.950, respectively. The results indicate a good model fit. Additionally, the root mean square residual (RMSR) and root mean square error of approximation (RMSEA) were calculated. The values of the measurement model are 0.058 and 0.068 for RMSR and RMSEA, respectively. The values satisfy the recommendation of [40]: RMSR < 0.10, RMSEA < 0.08. From the above results, it can be concluded that the proposed model has good fitness and can be developed to estimate the reliability and validity of the constructs in the model.

The composite reliability, average variance extracted, and Cronbach's alpha clarified the constructs' reliability and validity. The results are presented in Table 5. The composite reliability ranges from 0.678 to 0.938, which satisfies the composite reliability value of over 0.6. The average variance extracted values range from 0.512 to 0.791. All values are higher than the recommended value of 0.5 [41]. The values of Cronbach's alpha show an acceptance result above the threshold of 0.7 [42], except for the user interface (value = 0.677). In summary, the validity and reliability of the constructs were proven using variable statistics.

The regression weight and squared multiple correlations proved the correlation between the constructs and their corresponding variables. Table 6 presents the results. The regression weight for recognition is suggested to be greater than 0.5 for recognition as significant [41]. All values of the variables (ranging from 0.66 to 0.93) satisfy the suggestion ($p < 0.01$ for each item). In addition, the multiple correlations between the variables and their relevant constructs are over 0.40. The results show excellent correlations between the variables and constructs.

In conclusion, the measurement model and its constructs have adequate fitness, reliability, and validity. It is possible to develop a path analysis to clarify the constructs' coefficients in the proposed model.

**Table 5. Descriptive statistics of constructs.**

| Factor | Mean | Standard deviation | Composite reliability | Average variance extracted | Cronbach's alpha |
|---|---|---|---|---|---|
| PU | 3.36 | 1.00 | 0.938 | 0.791 | 0.936 |
| PE | 2.93 | 0.93 | 0.807 | 0.678 | 0.799 |
| ST | 2.92 | 0.92 | 0.794 | 0.659 | 0.789 |
| AT | 3.43 | 0.95 | 0.841 | 0.728 | 0.836 |
| IM | 3.58 | 0.96 | 0.783 | 0.547 | 0.773 |
| UI | 2.64 | 0.97 | 0.678 | 0.512 | 0.677 |

PU, perceived usefulness; PE, perceived enjoyment; ST, satisfaction; AT, attitude; IM, interactive media; UI, user interface.

**Table 6. Correlation results between the constructs and variables.**

|      | Standard regression weights | Squared multiple correlations |
|------|------|------|
| PU   |      |      |
| PU1  | 0.83 | 0.69 |
| PU2  | 0.93 | 0.86 |
| PU3  | 0.92 | 0.85 |
| PU4  | 0.87 | 0.76 |
| PE   |      |      |
| PE1  | 0.76 | 0.58 |
| PE2  | 0.88 | 0.77 |
| ST   |      |      |
| ST1  | 0.76 | 0.58 |
| ST2  | 0.86 | 0.74 |
| AT   |      |      |
| AT1  | 0.93 | 0.87 |
| AT2  | 0.77 | 0.59 |
| IM   |      |      |
| IM1  | 0.66 | 0.43 |
| IM2  | 0.78 | 0.61 |
| IM3  | 0.78 | 0.60 |
| UI   |      |      |
| UI1  | 0.74 | 0.55 |
| UI2  | 0.69 | 0.48 |

PU, perceived usefulness; PE, perceived enjoyment; ST, satisfaction; AT, attitude; IM, interactive media; UI, user interface.

The results of the path analysis are presented in Table 7. The results support H1 to H7 because of the significant paths ($p < 0.05$). The path from perceived enjoyment to satisfaction has the strongest relationship (value = 0.907). The lowest value of the path is from satisfaction to perceived usefulness, which is 0.394, and the other path coefficients range from 0.413 to 0.543. The variances in the constructs are shown in Fig 3. Satisfaction variable explains 41% of perceived usefulness, perceived usefulness explains perceived enjoyment with 46%, perceived enjoyment variable explains 88% of satisfaction, whereas perceived usefulness and perceived enjoyment explain 74% of attitude. In addition, attitude explains 25% and 29% of interactive media and user interface, respectively.

**Table 7. Path analysis results.**

|      | Path | Path Coefficient | Standard Error | Critical Ratio |
|------|------|------|------|------|
| H1 | PU → PE | 0.487 | 0.138 | 2.929** ($p = 0.003$) |
| H2 | ST → PU | 0.394 | 0.241 | 2.094* ($p = 0.036$) |
| H3 | PE → ST | 0.907 | 0.050 | 17.139*** |
| H4 | PU → AT | 0.413 | 0.044 | 8.231*** |
| H5 | PE → AT | 0.513 | 0.055 | 9.838*** |
| H6 | AT → IM | 0.543 | 0.051 | 10.289*** |
| H7 | AT → UI | 0.499 | 0.055 | 8.175*** |

*$p<0.05$, **$p<0.01$, ***$p<0.001$. PU, perceived usefulness; PE, perceived enjoyment; ST, satisfaction; AT, attitude; IM, interactive media; UI, user interface.

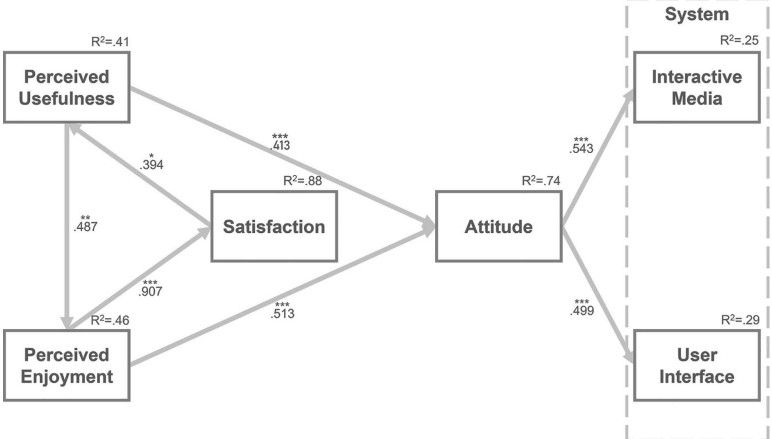

**Fig 3. Path analysis results.**

After clarifying the correlations between the constructs, linear regression analyses were used to describe the influence of the respondent's gender and age on the constructs (male = 1 and female = 2). The standard coefficients and t-values are presented in Table 8.

The results in Table 8 show that perceived usefulness, perceived enjoyment, and attitude are significantly influenced by gender ($p < 0.01$). These results partly prove H8. Regarding the influence of age, there are significant relationships among attitude, user interface, and age. These results demonstrate several features of H9. Meanwhile, the variance inflation factor (VIF) confirms no multicollinearity between the dependent variables [43].

Therefore, the relationships between demographics and constructs of the proposed model are explained as follows: gender has a significant relationship with perceived usefulness, perceived enjoyment, and attitude. Age has a substantial connection with attitude and the user interface.

## Discussions

### Modified TAM

This study proposed and evaluated a modified TAM to investigate driver acceptance of the DVIS. We pioneered the related effort of extending TAM with UX to study the DVIS, a newly emerging vehicle system. In this study, the reliability and validity of the proposed model and its constructs were proven, and the correlations between the constructs were clarified. In addition, we explored the influence of respondents' gender and age on their perceptions of the DVIS. The results supported the appropriateness of adopting the proposed model to investigate Japanese driver's perception of utilizing the DVIS.

### Perceived usefulness

The results on perceived usefulness show high reliability, supporting our proposal to modify the wording in perceived usefulness for investigating the DVIS. The information variables explain the perceived usefulness of DVIS. Our finding extends the relationship between perceived enjoyment and perceived usefulness in adopting TAM in vehicle systems, which supported [44].

### Perceived enjoyment

Perceived enjoyment has been proven to have a more decisive influence on attitude than does perceived ease of use in this study and supports the previous studies [45,46].

**Table 8. Hypotheses testing.**

|  | Beta | t-value | VIF | Tolerance | R² |
|---|---|---|---|---|---|
| PU |  |  |  |  |  |
| = ST | 0.595 | 17.614*** | 1.010 | 0.990 | 0.446 |
| + Gender | 0.235 | 6.886*** | 1.032 | 0.969 |  |
| + Age | -0.054 | -0.257 (*p* = 0.112) | 1.024 | 0.976 |  |
| PE |  |  |  |  |  |
| = PU | 0.650 | 17.475*** | 1.105 | 0.905 | 0.385 |
| + Gender | -0.135 | -3.623*** | 1.117 | 0.895 |  |
| + Age | 0.031 | 0.861 (*p* = 0.390) | 1.029 | 0.972 |  |
| ST |  |  |  |  |  |
| = PE | 0.764 | 26.418*** | 1.003 | 0.997 | 0.591 |
| + Gender | 0.052 | 1.778 (*p* = 0.076) | 1.027 | 0.974 |  |
| + Age | -0.015 | -0.503 (*p* = 0.615) | 1.024 | 0.977 |  |
| AT |  |  |  |  |  |
| = PU | 0.516 | 13.325*** | 1.793 | 0.558 | 0.590 |
| + PE | 0.312 | 8.442*** | 1.627 | 0.615 |  |
| + Gender | 0.087 | 2.808** (*p* = 0.005) | 1.147 | 0.872 |  |
| + Age | 0.072 | 2.445** (*p* = 0.015) | 1.031 | 0.970 |  |
| IM |  |  |  |  |  |
| = AT | 0.469 | 11.364*** | 1.067 | 0.937 | 0.218 |
| + Gender | -0.022 | -0.537 (*p* = 0.591) | 1.092 | 0.915 |  |
| + Age | 0.052 | 1.279 (*p* = 0.202) | 1.025 | 0.976 |  |
| UI |  |  |  |  |  |
| = AT | 0.344 | 7.954*** | 1.067 | 0.937 | 0.138 |
| + Gender | 0.005 | 0.122 (*p* = 0.903) | 1.092 | 0.915 |  |
| + Age | 0.139 | 3.270** (*p* = 0.001) | 1.025 | 0.976 |  |

*$p<0.05$, **$p<0.01$, ***$p<0.001$. Beta: standardized coefficient. PU, perceived usefulness; PE, perceived enjoyment; ST, satisfaction; AT, attitude; IM, interactive media; UI, user interface.

An interesting finding is that perceived enjoyment has a more substantial effect on attitude than perceived usefulness. Two reasons support this finding. One explanation is related to experience. If a driver is unfamiliar with a system at an early stage, they must follow the description of the system to utilize its functions. In this case, perceived usefulness is significant. However, when the driver acquires sufficient experience with the system's functions, the driver uses the system without instructions. Thus, the influence of perceived usefulness decreases, whereas the significance of perceived enjoyment increases. The second explanation concerns the DVIS attribute. Compared to a practical system, the driver would like to use DVIS functions to satisfy their hedonic requirements. For example, seats, lights, and climate can be adjusted using the DVIS to create a comfortable environment. In addition, they utilize DVIS to execute NDRTs (e.g., watching a movie or playing a game).

## Satisfaction

The results show that satisfaction is significantly correlated with perceived usefulness and perceived enjoyment. However, satisfaction has a positive relationship with perceived usefulness (H2), which is different from [47]. This difference is explained by the fact that a system

with higher satisfaction than other systems create a more profound impression on users and enables them to recognize it quickly.

Satisfaction has the highest coefficient for perceived enjoyment in the proposed model. This finding shows a robust connection between satisfaction and perceived enjoyment. However, Tamborini et al. [48] argued that satisfaction explains perceived enjoyment rather than its consequences of perceived enjoyment. We argue that satisfaction is a user's evaluation of the hedonic needs of a system. A driver should experience a system's functions and conclude their assessments after the experience. Thus, perceived enjoyment causes satisfaction.

Another point to discuss is that satisfaction does not influence attitude in the model. We consider that satisfaction indirectly affects attitude through perceived enjoyment because of the robust association between perceived enjoyment and satisfaction. It is unnecessary to connect satisfaction with attitude.

## Attitude

Consistent with H4 to H7, attitude is positively influenced by perceived usefulness and enjoyment, and it correlates with the system characteristics. This finding supports prior studies that attitude is more suitable than behavioral intention for studying users' perceptions of utilizing hedonic systems [16,37]. Meanwhile, the two questions on attitude show high reliability, which explains why these questions are suitable for investigating the cognitive and affective variables of attitude.

## System characteristics

From the analysis, it can be claimed that attitude influences two system characteristics (i.e., interactive media and user interface). However, the reliability results show that the values of the interactive media and user interface are not high. Meanwhile, attitude explains only 25% and 29% of interactive media and user interface, respectively. One explanation for these results is that some respondents may lack experience using interactive media (i.e., voice and gesture control). They need to imagine how to use these media to complete these variables in the questionnaire. Hence, the reliability of the variables is not high. Another explanation is the correlation between interactive media and user interface. Drivers should input their intentions into the system through interactive media and receive feedback through the user interface to complete the interaction process. The results can be improved by considering the potential relationship between interactive media and user interface.

## Demography

The results of the linear regression analysis reveal that gender (male = 1 and female = 2) significantly influences perceived usefulness (Beta = 0.235, $p < 0.01$), perceived enjoyment (Beta = -0.135, $p < 0.01$), and attitude (Beta = 0.087, $p < 0.01$). Compared with males, females showed less understanding and motivation to utilize the DVIS. However, they felt more comfortable with the functions and environments provided by these systems. Temple and Lips [49] supported female's negative attitude and low comprehension of the DVIS. This situation is related to females' anxiety about their abilities and skills to learn how to use information systems. Regarding female respondents' higher perception of perceived enjoyment than that of male respondents, we explain that perceived enjoyment is based on drivers' understanding of the systems' functions and environments. Some female drivers with less knowledge of the system may dismiss complex functions (e.g., adaptive cruise control) and concentrate on simple functions. Thus, their evaluations of perceived enjoyment are unilateral, which resulted in a higher assessment of perceived enjoyment.

The relationships between age and constructs of the proposed model are also worth discussing. Age has a positive influence on users' attitudes toward DVIS. However, this does not affect perceived usefulness and perceived enjoyment. This finding supports [50]. Regarding the finding that older drivers were unwilling to use the DVIS and disliked utilizing a user interface screen, we explain that this is because of the gaps between the elderly and younger groups, such as physiological, experiential, and cognitive factors [51]. Users' attitudes toward the DVIS change from positive to negative with decreasing cognitive and learning abilities in system functions. Older respondents lacked experience in receiving information on digital screens, leading them to dislike the digital user interface. Meanwhile, older people have already become familiar with conventional vehicle user interfaces. They preferred to receive information provided by mechanical parts rather than screens.

## Implication

This study provides several implications for designing a vehicle HMI to improve driver acceptance of systems.

First, the wording of the variables of perceived usefulness suggests that improving drivers' understanding of the system functions' information can enhance system comprehension and adoption. Vehicle manufacturers should organize related training lessons.

Second, the negative attitudes of the elderly toward the variables of attitude illustrate their conservative concept of using unfamiliar technologies (i.e., voice and gesture control), which require designing interfaces consistent with existing vehicle functions. For example, developing a consistent interface style and clear navigation flow.

Another practical implication concerns respondents' negative assessments of interactive media. This indicates that drivers prefer traditional interaction methods such as buttons and knobs. We suggest remaining conventional interaction methods that may improve driver safety and system usability.

Moreover, considering the correlations between age and user interface, a non-intrusive design [52] that utilizes peripheral attention areas for system notifications could minimize driver distractions and improve their accessibility.

## Limitations

**Respondent consideration**: one limitation of this study is the period of respondent consideration. One such example is perceived enjoyment. Davis et al. [53] argued that the definition of perceived enjoyment should explain its period. However, the cross-sectional survey design of this study limits the ability to specify the period of enjoyment, as it provides only a snapshot of respondent's opinions at a single time point.

**Actual system use**: in this study, we only investigated respondents' attitudes and intentions of the DVIS by their imagination rather than providing the system for respondents to experience. To solve these two limitations, we will design and evaluate a DVIS prototype to provide an actual experience to the participants. Then, provided the same questionnaire for them to answer.

**Perceived safety**: this questionnaire does not consider the perceived safety of executing NDRTs with a DVIS, which influences the driver's perceptions and attitudes toward the DVIS. Thus, we will develop an emergency braking function on a designed DVIS prototype and conduct a related experiment, expecting to improve the driver's safety for him/her to accept the DVIS.

**Regional characteristics of demographics**: the diverse economic situations between developing and developed countries should be considered when exploring customers' perceptions

of the system and technology [54,55]. In the future, we intend to explore the regional factors (e.g., culture, religion, and country development) that influence drivers' acceptance of the DVIS by translating the questionnaire and distributing it in different countries.

**The correlation between satisfaction and demographics**: further studies are needed to confirm the correlation between satisfaction and demographics. One method is to design new variables of satisfaction to discuss the different driver's technology anxiety levels (for the investigation of age) and interaction styles (for the investigation of gender).

## Conclusions

This study proposed and validated a modified technology acceptance model to investigate Japanese drivers' acceptance of driver-vehicle interaction systems.

Our method introduced user experience concepts into the technology acceptance model. It was proven to be reliable and valid in investigating driver-vehicle interaction system acceptance. The six constructs (i.e., perceived usefulness, perceived enjoyment, satisfaction, attitude, interactive media, and user interface) showed good reliability and correlations in the research model. In addition, gender and age significantly influenced driver's perceptions and attitudes toward driver-vehicle interaction systems.

We will continue developing research to conduct an experiment to test a designed prototype of a driver-vehicle interaction system, expecting to evaluate their acceptance of the system by our modified technology acceptance model.

## Supporting information

**S1 File.  Blue&Me information.**
(DOCX)

**S1 Table.  Questionnaire content.**
(DOCX)

## Acknowledgment

We thank our colleagues from the Laboratory for Cognitive Systems Science, University of Tsukuba, for their opinions on model design and their checking work for the questionnaire.

## Author contributions

**Conceptualization:** Yichen Dong, Makoto Itoh.

**Data curation:** Yichen Dong.

**Formal analysis:** Yichen Dong.

**Funding acquisition:** Makoto Itoh.

**Investigation:** Yichen Dong.

**Methodology:** Yichen Dong.

**Project administration:** Makoto Itoh.

**Resources:** Yichen Dong.

**Software:** Yichen Dong.

**Supervision:** Makoto Itoh.

**Validation:** Yichen Dong.

**Visualization:** Yichen Dong.

**Writing – original draft:** Yichen Dong.

**Writing – review & editing:** Yichen Dong, Makoto Itoh.

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
