## [Decision Letter · Decision Letter 0]

29 Oct 2024

PONE-D-24-38763A modification of technology acceptance model for investigating driver-vehicle interaction systems usagePLOS ONE

Dear Dr. Dong,

Thank you for submitting your manuscript to PLOS ONE. After careful consideration, we feel that it has merit but does not fully meet PLOS ONE’s publication criteria as it currently stands. Therefore, we invite you to submit a revised version of the manuscript that addresses the points raised during the review process.

**ACADEMIC EDITOR: Based on the reviewers comments i suggest you to cite the following articles and compare your findings to enhance it scope. **

https://doi.org/10.1145/3498851.3498992https://doi.org/10.1007/s11116-024-10515-3 ==============================

We look forward to receiving your revised manuscript.

Kind regards,

Sohaib Mustafa, Ph.D.

Academic Editor

PLOS ONE

2. We note that you have indicated that there are restrictions to data sharing for this study. PLOS only allows data to be available upon request if there are legal or ethical restrictions on sharing data publicly. For more information on unacceptable data access restrictions, please see http://journals.plos.org/plosone/s/data-availability#loc-unacceptable-data-access-restrictions. Before we proceed with your manuscript, please address the following prompts: a) If there are ethical or legal restrictions on sharing a de-identified data set, please explain them in detail (e.g., data contain potentially identifying or sensitive patient information, data are owned by a third-party organization, etc.) and who has imposed them (e.g., a Research Ethics Committee or Institutional Review Board, etc.). Please also provide contact information for a data access committee, ethics committee, or other institutional body to which data requests may be sent. b) If there are no restrictions, please upload the minimal anonymized data set necessary to replicate your study findings to a stable, public repository and provide us with the relevant URLs, DOIs, or accession numbers. For a list of recommended repositories, please see https://journals.plos.org/plosone/s/recommended-repositories. You also have the option of uploading the data as Supporting Information files, but we would recommend depositing data directly to a data repository if possible. We will update your Data Availability statement on your behalf to reflect the information you provide.

Additional Editor Comments (if provided):

Reviewers' comments:

Reviewer's Responses to Questions

**Comments to the Author**

1. Is the manuscript technically sound, and do the data support the conclusions?

Reviewer #1: Partly

Reviewer #2: Partly

2. Has the statistical analysis been performed appropriately and rigorously? 

Reviewer #1: Yes

Reviewer #2: Yes

3. Have the authors made all data underlying the findings in their manuscript fully available?

Reviewer #1: Yes

Reviewer #2: Yes

4. Is the manuscript presented in an intelligible fashion and written in standard English?

Reviewer #1: Yes

Reviewer #2: No

5. Review Comments to the Author

Reviewer #1: Dear Author(s)

The paper makes a valuable contribution by investigating respondents' acceptance of driver-vehicle interaction systems through a questionnaire survey and explaining the proposed model constructs, which include perceived usefulness, perceived enjoyment, satisfaction, attitude, interactive media, and user interface. While the article is well-known and has a solid scientific structure, I suggest the following clarifications and additions (Major revisions) before it is considered for publication:

Abstract:

It is necessary to include the following items in this section:

- In the “Abstract” section; no information has been provided regarding the type of study, the number of participants, the tools/questionnaires used, and so on. The research findings should be presented quantitatively in the "Abstract" section. Additionally, a general conclusion from the study should be provided as the "Conclusion".

- Please state the significance level (p-value) in which this analysis was performed in the results section.

Introduction:

- There is no need to introduce the three determinant constructs of TAM (as illustrated in Fig 1) in the “Introduction” section. It is recommended to present it in the "Materials and Methods" section under a subsection titled “Technical acceptance model”.

- The authors need to revise and rewrite the text to improve coherence between the paragraphs.

Material and methods:

- In the “Materials and Methods” section, the authors should present the model constructs′ explanations more concisely and avoid providing irrelevant details. It is recommended to present these constructs in a table along with their definitions and corresponding references. This approach will improve the clarity of the components, enhance readability for the reader, and ultimately make the text more concise.

- Figure 1 and its explanation should be presented in this section, not in the "Introduction" section.

- Please organize the study hypotheses into a table.

- In the “Materials and Methods” section; the authors need to provide information on the validity and reliability of the questionnaire. Additionally, please specify whether the questionnaire was distributed to participants in English or Japanese. If it was translated into Japanese, was its validity verified?

- In “Data analysis” subsection; Was the assumption of normality of the data evaluated? If yes, with what test?

- In “Data analysis” subsection; Were outliers checked in the study? At what confidence level was the data analysis conducted?

Results: Good

Discussion:

- The authors have listed the study's limitations separately under the subsection “Limitations”. It is essential to include this information as a paragraph at the end of the “Discussion” section. It is necessary to express the strengths and limitations of the present study more clearly. Can the authors state more limitations for the present study? Also, the authors should provide their suggestions for future studies (if any). I believe the stated limitations are unclear and may be difficult for the reader to understand.

Conclusions:

- Authors are expected to present a complete and adequate conclusion of their study in this section. In the “Conclusion” section, authors should present a general conclusion of the study, not the study findings.

- Some of the study's limitations are mentioned in the “Conclusion” section. These limitations should be consolidated and presented together in the final paragraph of the “Discussion” section.

Reviewer #2: 1. The topic is very interesting and applicable to present day scenario.

2. However, as originated in Japan, it has considered road features and driving conditions of Japan only where automated driving is very much accepted. It has not considered the fact that how much 'automated driving is acceptable in global scenario' or 'the degree of automation that can be considered safe in global scenario'. This is considered important because for each person, he himself is the safest driver. So in such crucial aspect how much can we depend on a 'machine for decision making' is a million dollar question. Please go through following paper :

Michael A. Nees, 2019. Safer than the average human driver (who is less safe than me)? Examining a popular

safety benchmark for self-driving cars. Journal of Safety Research, Volume 69, June 2019, Pages 61-68.

I would request that, as this paper is a questionnaire based study, the author could discuss "the degree of automation accepted" and "Safety of Driver and Public" before discussing the HMI of Driver-Vehicle Interaction. These two aspects would also influence the user experience and perceived usefulness under Technology Acceptance Model. Authors would appreciate the fact that if the Drivers are completely satisfied about safety aspects of automated driving, then only they will be able to divert their attention (without increasing the cognitive workload) to execute Non-Driving Related Tasks (NDRT), like, playing a game or watching a movie (Refer to the two enclosed files).

3. Sentences should never start with abbreviations (e.g., refer paragraph starting at line no. 68). Kindly correct it throughout the manuscript.

4. The sentences have to be meaningful. Please correct it throughout the manuscript. For example, refer line 88, "Several studies.. introduced TAM ... systems." Instead of the word 'introduced' the word 'used' would be giving the meaning in better way. Logically several studies cannot introduce something. It is understood that for the first time something is introduced and after that it is used.

5. Please use simple and easily understandable English. Please do not use words like 'adhibition' line no. 134. kindly rephrase this sentence and correct others throughout the manuscript.

6. Kindly correct grammar throughout the manuscript. For example, refer line no. 390, remove 'the' in "On the one hand...". The manuscript appears to be written in short-cut language and is devoid of proper sentence construction at a number of instances.

7. Table 5 appears to be too long.

8. Conclusion cannot be two pages long. Kindly condense it in 3-4 bulleted sentences.

9. Enclosed figures are neither legible nor readable. Please attach them in clear picture format.

6. PLOS authors have the option to publish the peer review history of their article (what does this mean? ). If published, this will include your full peer review and any attached files.

**Do you want your identity to be public for this peer review?** For information about this choice, including consent withdrawal, please see our Privacy Policy .

Reviewer #1: **Yes: ** Sajjad Rostamzadeh

Reviewer #2: **Yes: ** Dr Deepti Majumdar

---

## [Author Response · Author response to Decision Letter 1]

21 Nov 2024

Dear Editor,

Thank you for your questions and comments.

We have carefully addressed your concerns as shown in this response letter and revised the manuscript.

Please check the “Revised Manuscript with Track Changes” for tracking our revisions. The major changes made in the revised manuscript have been highlighted in yellow (the deleted contents and revisions made on language problems are not included). We have also uploaded a file called “Revised Manuscript with the Whole Track Changes” which covers all the change histories.

Based on the reviewers’ comments I suggest you to cite the following articles and compare your findings to enhance its scope.

Reply: Thank you for the references, we have carefully read the references and added the citations in the paper to enhance the scope of our discussion of perceived usefulness and enjoyment. Please check lines 156 to 159, 171 to 173, and 543 to 547.

Journal requirements

Reply: We have checked and modified the manuscript and file names to meet PLOS ONE's style requirements following the links.

2. We note that you have indicated that there are restrictions to data sharing for this study. PLOS only allows data to be available upon request if there are legal or ethical restrictions on sharing data publicly.

Reply: We have uploaded the data on the “Dryad”, DOI: 10.5061/dryad.4qrfj6qm5.

Please let us know if you have any questions or further feedback.

I hope our revisions can satisfy your comments and questions.

Sincerely yours,

Dong and Itoh 

Dear Reviewer #1,

Thank you for your questions and comments.

We have carefully addressed your concerns as shown in this response letter and revised the manuscript.

Please check the “Revised Manuscript with Track Changes” for tracking our revisions. The major changes made in the revised manuscript have been highlighted in yellow (the deleted contents and revisions made on language problems are not included). We have also uploaded a file called “Revised Manuscript with the Whole Track Changes” which covers all the change histories.

Abstract:

It is necessary to include the following items in this section:

- In the “Abstract” section; no information has been provided regarding the type of study, the number of participants, the tools/questionnaires used, and so on. The research findings should be presented quantitatively in the "Abstract" section. Additionally, a general conclusion from the study should be provided as the "Conclusion".

Reply: We have added the type of study, the details of the participants, and the information about the questionnaire used in the “Abstract” from lines 30 to 37. The general conclusion of the study has been added from lines 48 to 51.

- Please state the significance level (p-value) in which this analysis was performed in the results section.

Reply: We have stated the significant level at lines 40, 44, and 46.

Introduction:

- There is no need to introduce the three determinant constructs of TAM (as illustrated in Fig 1) in the “Introduction” section. It is recommended to present it in the "Materials and Methods" section under a subsection titled “Technical acceptance model”.

Reply: We have moved the introduction of TAM from the “Introduction” section to the “Materials and methods” section where a subsection called “TAM” was created at line 91.

- The authors need to revise and rewrite the text to improve coherence between the paragraphs.

Reply: We have deleted some parts of “Introduction” and added a paragraph from lines 84 to 87 to improve the coherence between the sections “Introduction” and “Materials and methods”.

Material and methods:

- In the “Materials and Methods” section, the authors should present the model constructs′ explanations more concisely and avoid providing irrelevant details. It is recommended to present these constructs in a table along with their definitions and corresponding references. This approach will improve the clarity of the components, enhance readability for the reader, and ultimately make the text more concise.

Reply: We have deleted the irrelevant details and modified the sentences to improve the coherence between the paragraphs. The modifications are highlighted in yellow. In addition, a section named “Proposed model” was added to include the subsections of the model constructs and added the general introduction from lines 150 to 153. Regarding the table of the constructs, we have introduced the definitions of the constructs and their references in Table 1 at line 238.

- Figure 1 and its explanation should be presented in this section, not in the "Introduction" section.

Reply: Figure 1 has been moved to the “TAM” subsection at line 101.

- Please organize the study hypotheses into a table.

Reply: We have added a new subsection named “Hypotheses” and put the study hypotheses into Table 2 at line 256.

- In the “Materials and Methods” section; the authors need to provide information on the validity and reliability of the questionnaire. Additionally, please specify whether the questionnaire was distributed to participants in English or Japanese. If it was translated into Japanese, was its validity verified?

Reply: Because this study is a continuation of a previous study [19] on questionnaire design, we had proved the questionnaire’s validity and reliability with exploratory factor analysis in that study. Regarding the language of the questionnaire, we had used Japanese language translated from Chinese language. We had invited a bilingualist who has expertise in these two languages to check the contents of the questionnaire. We have added these explanations from lines 278 to 283 in the subsection “Questionnaire design”.

- In “Data analysis” subsection; Was the assumption of normality of the data evaluated? If yes, with what test?

Reply: Because the data is based on the questionnaire survey, we had used the absolute value of skewness and kurtosis to check the normality. We have added the values of skewness and kurtosis in Table 3. Please check the sentences from lines 297 to 300 and Table 3 at line 301. In addition, we have changed the subsection title to “Measurement and data analysis” which might be more suitable.

- In “Data analysis” subsection; Were outliers checked in the study? At what confidence level was the data analysis conducted?

Reply: We have added the strategy of deleting invalid data from lines 294 to 297 and added the confidence level from lines 306 to 307.

Discussion:

- The authors have listed the study's limitations separately under the subsection “Limitations”. It is essential to include this information as a paragraph at the end of the “Discussion” section. It is necessary to express the strengths and limitations of the present study more clearly. Can the authors state more limitations for the present study? Also, the authors should provide their suggestions for future studies (if any). I believe the stated limitations are unclear and may be difficult for the reader to understand.

Reply: We have rewritten the subsection of “Limitations” to make it clearer for reading. We have added and discussed 5 limitations in the “Limitations” subsection: the period of respondent’s consideration, actual system use, perceived safety, regional characteristics, and the correlation between satisfaction and demographics. For each limitation, we have explained our continued works and future plans. Please check the “Limitations” subsection from lines 574 to 604.

Conclusions:

- Authors are expected to present a complete and adequate conclusion of their study in this section. In the “Conclusion” section, authors should present a general conclusion of the study, not the study findings.

Reply: We have rewritten the section “Conclusion” to provide general conclusion. Please check the section “Conclusion” from lines 606 to 621.

- Some of the study's limitations are mentioned in the “Conclusion” section. These limitations should be consolidated and presented together in the final paragraph of the “Discussion” section.

Rely: We have merged these sentences into the subsection of “Limitations”.

Please let us know if you have any questions or further feedback.

I hope our revisions can satisfy your comments and questions.

Sincerely yours,

Dong and Itoh

Dear Reviewer #2,

Thank you for your questions and comments.

We have carefully addressed your concerns as shown in this response letter and revised the manuscript.

Please check the “Revised Manuscript with Track Changes” for tracking our revisions. The major changes made in the revised manuscript have been highlighted in yellow (the deleted contents and revisions made on language problems are not included). We have also uploaded a file called “Revised Manuscript with the Whole Track Changes” which covers all the change histories.

1. The topic is very interesting and applicable to present day scenario.

Reply: Thank you for your interest in this paper.

2. I would request that, as this paper is a questionnaire based study, the author could discuss "the degree of automation accepted" and "Safety of Driver and Public" before discussing the HMI of Driver-Vehicle Interaction. These two aspects would also influence the user experience and perceived usefulness under Technology Acceptance Model. Authors would appreciate the fact that if the Drivers are completely satisfied about safety aspects of automated driving, then only they will be able to divert their attention (without increasing the cognitive workload) to execute Non-Driving Related Tasks (NDRT), like, playing a game or watching a movie (Refer to the two enclosed files).

Reply: Thank you for the attached papers that inspired our revisions. We agreed with your idea that the paper should discuss safety as well as the degree of automation accepted. Because we are researching the safety, usability, and user acceptance of our designed driver-vehicle interface prototype in the continued work of this paper currently. Thus, we have added the discussion of safety in the section of “Introduction” from lines 54 to 66 and discussed the limitation of safety in the subsection of “Limitations” from lines 587 to 592. Please check these paragraphs.

3. Sentences should never start with abbreviations (e.g., refer paragraph starting at line no. 68). Kindly correct it throughout the manuscript.

Reply: We have corrected the problem. We have also checked the manuscript to correct similar problems.

4. The sentences have to be meaningful. Please correct it throughout the manuscript. For example, refer line 88, "Several studies.. introduced TAM ... systems." Instead of the word 'introduced' the word 'used' would be giving the meaning in better way. Logically several studies cannot introduce something. It is understood that for the first time something is introduced and after that it is used.

Reply: The word was changed to “used”, please check line 105. We have also checked the manuscript to correct similar problems.

5. Please use simple and easily understandable English. Please do not use words like 'adhibition' line no. 134. kindly rephrase this sentence and correct others throughout the manuscript.

Reply: The word was changed to “development”, please check line 146. We have also checked the manuscript for similar problems.

6. Kindly correct grammar throughout the manuscript. For example, refer line no. 390, remove 'the' in "On the one hand...". The manuscript appears to be written in short-cut language and is devoid of proper sentence construction at a number of instances.

Reply: We have deleted these words, please check line 337. We have also done the proofread to avoid similar problems.

7. Table 5 appears to be too long.

Reply: To shorten the tables, we have used abbreviations and reduced the size of the words. Please check the table at line 391.

8. Conclusion cannot be two pages long. Kindly condense it in 3-4 bulleted sentences.

Reply: We have rewritten the conclusion by condensing it. Please check the conclusion from lines 606 to 621.

9. Enclosed figures are neither legible nor readable. Please attach them in clear picture format.

Reply: We have adjusted and reuploaded the figures, please check them.

Please let us know if you have any questions or further feedback.

I hope our revisions can satisfy your comments and questions.

Sincerely yours,

Dong and Itoh

---

## [Decision Letter · Decision Letter 1]

4 Mar 2025

PONE-D-24-38763R1A modification of technology acceptance model for investigating driver-vehicle interaction systems usagePLOS ONE

Dear Dr. Dong,

Thank you for submitting your manuscript to PLOS ONE. After careful consideration, we feel that it has merit but does not fully meet PLOS ONE’s publication criteria as it currently stands. Therefore, we invite you to submit a revised version of the manuscript that addresses the points raised during the review process.

We look forward to receiving your revised manuscript.

Kind regards,

Sohaib Mustafa, Ph.D.

Academic Editor

PLOS ONE

Journal Requirements:

Reviewers' comments:

Reviewer's Responses to Questions

**Comments to the Author**

1. If the authors have adequately addressed your comments raised in a previous round of review and you feel that this manuscript is now acceptable for publication, you may indicate that here to bypass the “Comments to the Author” section, enter your conflict of interest statement in the “Confidential to Editor” section, and submit your "Accept" recommendation.

Reviewer #1: All comments have been addressed

Reviewer #2: (No Response)

2. Is the manuscript technically sound, and do the data support the conclusions?

Reviewer #1: Yes

Reviewer #2: Partly

3. Has the statistical analysis been performed appropriately and rigorously? 

Reviewer #1: Yes

Reviewer #2: No

4. Have the authors made all data underlying the findings in their manuscript fully available?

Reviewer #1: Yes

Reviewer #2: Yes

5. Is the manuscript presented in an intelligible fashion and written in standard English?

Reviewer #1: Yes

Reviewer #2: No

6. Review Comments to the Author

Reviewer #1: (No Response)

Reviewer #2: 1.Abstracts : Kindly provide quantified and analyzed data that proves this observation (both in Abstract and main text) "Male drivers had higher perceived usefulness and more positive attitudes than female drivers".

2. Introduction : Improved and acceptable.

3. Materials & Methods :

i) Lines 145 - 147 : This sentence is not meaningful and seems to be incomplete. Kindly re-frame it. : "Prior studies on extending TAM support its modifications. The constructs' explanations and their correlations with other constructs are presented in the following hypotheses"

ii) Hypothesis is not clearly written. Kindly write the hypothesis in to-the-point and crisp manner.

iii) Research model and theories should not be discussed in Materials & Method Section. Please transfer chunk of the material to Discussion section. It is not clear that which methodology has been used to develop model and how the model is supposed to behave. Why a subheading of Methodology is being introduced under Materials and Methods section

4. Results : All Tables still need formatting.

5. Discussion Section : Please remove the repetition of text between Material & Methods and Discussion sections.

6. Please give precise implications of study as per the results obtained after questionnaire data analysis. It seems a bit jumbled up.

7. Length of the Limitation and Conclusion section are a bit lengthy, it is suggested bulleted points may please be given.

7. PLOS authors have the option to publish the peer review history of their article (what does this mean? ). If published, this will include your full peer review and any attached files.

**Do you want your identity to be public for this peer review?** For information about this choice, including consent withdrawal, please see our Privacy Policy .

Reviewer #1: **Yes: ** Sajjad Rostamzadeh

Reviewer #2: **Yes: ** Dr Deepti Majumdar

---

## [Author Response · Author response to Decision Letter 2]

10 Mar 2025

Dear Editor,

We have carefully addressed the reviewers’ comments as shown in this response letter and revised the manuscript.

Please check the “March Revised Manuscript with Track Changes” for tracking our revisions. The major changes made in the revised manuscript have been highlighted in yellow (the deleted contents and revisions made on language problems are not included). We have also uploaded a file called “March Revised Manuscript with the Whole Track Changes” which covers all the change histories.

Journal Requirements:

We have checked the reference list, and it is complete and correct. Meanwhile, all the references are not retracted.

Please let us know if you have any questions or further feedback.

Sincerely yours,

Dong and Itoh 

Dear Reviewer #2,

Thank you for your questions and comments.

First, sorry for not mentioning there are two manuscripts in the previous submission pdf: page 13/192 to 63/192 was the original manuscript without revision, and page 64/192 to 113/192 was the revised manuscript with the first-round revision. Sorry again for causing you any confusion in the review. Please check the download link on the top right-hand side of the manuscript's first page: March revision.docx for the newest version of this manuscript.

We have carefully addressed your concerns as shown in this response letter and revised the manuscript.

Please check the “March Revised Manuscript with Track Changes” for tracking. The major changes made in the revised manuscript are highlighted in yellow (the deleted contents and revisions made on language problems are not included). We also uploaded a file called “March Revised Manuscript with the Whole Track Changes” which covers all the change histories.

Abstracts:

1. Kindly provide quantified and analyzed data that proves this observation (both in Abstract and main text) "Male drivers had higher perceived usefulness and more positive attitudes than female drivers".

Reply: We have added the analysis results for the gender and age influence on the model’s constructs in both Abstract and Discussion sections. Please check lines 42 to 50 and 493 to 496.

Materials & Methods:

1. Lines 145 - 147: This sentence is not meaningful and seems to be incomplete. Kindly re-frame it.: "Prior studies on extending TAM support its modifications. The constructs' explanations and their correlations with other constructs are presented in the following hypotheses"

Reply: Sorry for the confusion of the different versions of the manuscript. This sentence was cut in the previous revision version.

2. Hypothesis is not clearly written. Kindly write the hypothesis in to-the-point and crisp manner.

Reply: Sorry about the confusion of the different versions of the manuscript. We had concluded the hypotheses together in Table 2 and added a new subsection named “Hypotheses” in the previous revision. Please check this section from line 256.

3. Research model and theories should not be discussed in Materials & Method Section. Please transfer chunk of the material to Discussion section. It is not clear that which methodology has been used to develop model and how the model is supposed to behave. Why a subheading of Methodology is being introduced under Materials and Methods section

Reply: Sorry about the confusion due to different versions of the manuscript. We had deleted the irrelevant details and modified the sentences to improve the coherence between the paragraphs in the previous version. In addition, we had added Table 1, Table 2, and Figure 2 to explain the development of model construction and the supports. Please check lines 242 to 244 and 256 to 263. We checked that Methodology is a heading equivalent to Materials and Methods section. Please check the table of contents.

Results:

1. All Tables still need formatting.

Reply: Sorry for the confusion of the different versions of the manuscript. We had used abbreviations and reduced the size of the words of the tables in the previous revision version. In addition, we reduced the line spaces of the tables this time. Please check lines 305, 339, 351, 364, 383, and 395 for these tables.

Discussion:

1. Please remove the repetition of text between Material & Methods and Discussion sections.

Reply: We deleted the irrelevant details and modified the sentences to show the differences between Material & Methods and Discussion sections. Please check this section from line 409.

2. Please give precise implications of study as per the results obtained after questionnaire data analysis. It seems a bit jumbled up.

Reply: We reduced the sentences in Implication subsection to make it clearer for reading. Please check this subsection from line 525.

3. Length of the Limitation and Conclusion section are a bit lengthy, it is suggested bulleted points may please be given.

Reply: Sorry for the confusion of the different versions of the manuscript. We had reduced the sentences in the sections of Limitation and Conclusion in the previous revision version. In addition, we highlighted each limitation to make it clearer to read now. Please check the sections from line 547.

Please let us know if you have any questions or further feedback. I hope our revisions can satisfy your comments and questions. Looking forward to your timely response.

Sincerely yours,

Dong and Itoh

---

## [Decision Letter · Decision Letter 2]

18 Mar 2025

A modification of technology acceptance model for investigating driver-vehicle interaction systems usage

PONE-D-24-38763R2

Dear Dr. Dong,

We’re pleased to inform you that your manuscript has been judged scientifically suitable for publication and will be formally accepted for publication once it meets all outstanding technical requirements.

Kind regards,

Sohaib Mustafa, Ph.D.

Academic Editor

PLOS ONE

Additional Editor Comments (optional):

Reviewers' comments:

Reviewer's Responses to Questions

**Comments to the Author**

1. If the authors have adequately addressed your comments raised in a previous round of review and you feel that this manuscript is now acceptable for publication, you may indicate that here to bypass the “Comments to the Author” section, enter your conflict of interest statement in the “Confidential to Editor” section, and submit your "Accept" recommendation.

Reviewer #2: All comments have been addressed

2. Is the manuscript technically sound, and do the data support the conclusions?

Reviewer #2: Yes

3. Has the statistical analysis been performed appropriately and rigorously? 

Reviewer #2: Yes

4. Have the authors made all data underlying the findings in their manuscript fully available?

Reviewer #2: Yes

5. Is the manuscript presented in an intelligible fashion and written in standard English?

Reviewer #2: Yes

6. Review Comments to the Author

Reviewer #2: Satisfied with authors' revision of the manuscript. Can be accepted for publication in the present form.

7. PLOS authors have the option to publish the peer review history of their article (what does this mean? ). If published, this will include your full peer review and any attached files.

**Do you want your identity to be public for this peer review?** For information about this choice, including consent withdrawal, please see our Privacy Policy .

Reviewer #2: **Yes: ** Dr Deepti Majumdar

---

## [Editor Report · Acceptance letter]

PONE-D-24-38763R2

PLOS ONE

Dear Dr. Dong,

I'm pleased to inform you that your manuscript has been deemed suitable for publication in PLOS ONE. Congratulations! Your manuscript is now being handed over to our production team.

Kind regards,

on behalf of

Dr. Sohaib Mustafa

Academic Editor

PLOS ONE